# Assessment and Risk Prediction of Chronic Kidney Disease and Kidney Fibrosis Using Non-Invasive Biomarkers

**DOI:** 10.3390/ijms25073678

**Published:** 2024-03-26

**Authors:** Harald Rupprecht, Lorenzo Catanese, Kerstin Amann, Felicitas E. Hengel, Tobias B. Huber, Agnieszka Latosinska, Maja T. Lindenmeyer, Harald Mischak, Justyna Siwy, Ralph Wendt, Joachim Beige

**Affiliations:** 1Department of Nephrology, Angiology and Rheumatology, Klinikum Bayreuth GmbH, 95445 Bayreuth, Germany; harald.rupprecht@klinikum-bayreuth.de (H.R.); lorenzoriccardo.catanese@gmail.com (L.C.); 2Department of Nephrology, Medizincampus Oberfranken, Friedrich-Alexander-University Erlangen-Nürnberg, 91054 Erlangen, Germany; 3Kuratorium for Dialysis and Transplantation (KfH) Bayreuth, 95445 Bayreuth, Germany; 4Department of Nephropathology, Institute of Pathology, Friedrich-Alexander-University Erlangen-Nürnberg, 91054 Erlangen, Germany; kerstin.amann@uk-erlangen.de; 5III Department of Medicine, University Medical Center Hamburg-Eppendorf, 20251 Hamburg, Germany; f.hengel@uke.de (F.E.H.); t.huber@uke.de (T.B.H.); m.lindenmeyer@uke.de (M.T.L.); 6Hamburg Center for Kidney Health (HCKH), University Medical Center Hamburg Eppendorf, 20246 Hamburg, Germany; 7Mosaiques Diagnostics GmbH, 30659 Hannover, Germany; latosinska@mosaiques-diagnostics.com (A.L.); mischak@mosaiques-diagnostics.com (H.M.); siwy@mosaiques-diagnostics.com (J.S.); 8Department of Nephrology, Hospital St. Georg, 04129 Leipzig, Germany; ralph.wendt@sanktgeorg.de; 9Kuratorium for Dialysis and Transplantation (KfH) Renal Unit, Hospital St. Georg, 04129 Leipzig, Germany; 10Department of Internal Medicine II, Martin-Luther-University Halle/Wittenberg, 06108 Halle (Saale), Germany

**Keywords:** biomarker, kidney disease, response prediction, progression, fibrosis

## Abstract

Effective management of chronic kidney disease (CKD), a major health problem worldwide, requires accurate and timely diagnosis, prognosis of progression, assessment of therapeutic efficacy, and, ideally, prediction of drug response. Multiple biomarkers and algorithms for evaluating specific aspects of CKD have been proposed in the literature, many of which are based on a small number of samples. Based on the evidence presented in relevant studies, a comprehensive overview of the different biomarkers applicable for clinical implementation is lacking. This review aims to compile information on the non-invasive diagnostic, prognostic, and predictive biomarkers currently available for the management of CKD and provide guidance on the application of these biomarkers. We specifically focus on biomarkers that have demonstrated added value in prospective studies or those based on prospectively collected samples including at least 100 subjects. Published data demonstrate that several valid non-invasive biomarkers of potential value in the management of CKD are currently available.

## 1. Introduction

Early detection of chronic kidney disease (CKD) and prognosis of disease progression are highly relevant in the effective management of CKD, as well as when considering health economic issues [1,2]. The conventional biomarkers, estimated glomerular filtration rate (eGFR) and urinary albumin to creatinine ratio (UACR), have apparent shortcomings in this context. They are the consequence of kidney damage but are generally not connected with the molecular changes responsible for the initiation and progression of CKD [3,4]. There is an urgent need for sensitive and specific non-invasive biomarkers to improve patient management (Figure 1).

The biomarkers should achieve the following goals:Diagnosis of kidney disease in the early stages: Using conventional diagnostic criteria of eGFR < 60 mL/min/1.73 m^2^ and/or UACR > 30 or 300 mg/g creatinine, a significant proportion of patients with early and progressing CKD remain undetected and preventive measures are not applied. This is particularly important in the growing population of elderly patients, in whom GFR thresholds are even less consistent than those in the younger population;Prediction of the risk of CKD progression: Identifying patients at a high risk of progression is of utmost importance in order to offer adequate therapy. Likewise, identifying patients who are not at risk of progression would avoid unnecessary therapeutic interventions;Estimation of the degree of kidney fibrosis (KF): Interstitial fibrosis and tubular atrophy (IFTA) is the histological marker with the strongest predictive power for progressive loss of kidney function. It would be desirable to predict the degree of IFTA using a non-invasive biomarker;Prediction of therapy response: Differentiating responders from non-responders at baseline would be an essential step towards personalized nephrology and individualized therapy;Monitoring the effects of therapeutic interventions: Non-invasive sample collection enables multiple samplings throughout the course of the disease, allowing for the monitoring of therapeutic success.

We reviewed the current literature on non-invasive biomarkers to detect CKD, assess the progression of CKD and the degree of KF, as well as predict the response to specific therapeutic interventions, with emphasis on clinical applicability. As a basis, we investigated PubMed using the following search terms: (((CKD[Title/Abstract] OR “chronic kidney disease”[Title/Abstract]) NOT (“acute kidney injury” OR AKI[Title/Abstract])) AND (biomarker[Title/Abstract] OR classifier[Title/Abstract])) AND ((fibrosis[Title/Abstract]) OR (predict* AND (therap* OR treatment) AND respon*[Title/Abstract]) OR (“for early detection” OR “for early diagnosis”[Title/Abstract])). This search returned 197 manuscripts published from 2018 to 2023 which were independently assessed by at least two co-authors for their relevance to the context of this review.

## 2. Non-Invasive Biomarkers for CKD Detection and Assessment of Risk of Progression

### 2.1. Routine Clinical Markers

Currently, GFR estimates based on measurement of serum creatinine and UACR are most frequently used to detect and classify CKD stages, with the shortcomings described above. A further major shortcoming of the UACR is its high biological variability, with a coefficient of variation (CV) exceeding 30% [5,6]. CVs for eGFR are reported to be substantially lower, in the range of 12% [7], which still implies a substantial element of uncertainty when evaluating disease progression based on changes in eGFR. Both creatinine- and cystatin C-based GFR estimations enable CKD detection with increased confidence at advanced disease stages. However, the two estimates do not show a perfect agreement. To predict the consequences of CKD, end-stage kidney disease (ESKD), cardiovascular manifestations, or mortality, eGFR_cystatin C_ appears superior to eGFR_creatinine_ [8,9,10]. One hypothesis for the superiority of eGFR_cystatin C_ is based on the assumption that cystatin C (13.3 kDa) is superior to creatinine (0.113 kDa) in identifying disruption in the filtration process in the pore model of glomerular filtration, predicting a reduced filtration of proteins or peptides in the 5–30 kDa range [11]. The term “shrunken pore syndrome” (SPS) has been proposed for the associated pathophysiological mechanism, diagnosed by the ratio of eGFR_cystatin C_ vs. eGFR_creatinine_ [12]. Most individuals display a ratio between 0.9 and 1. However, in a study of 1349 randomly selected subjects, 8% showed a ratio < 0.6 in the absence of factors that perturbed the use of cystatin C or creatinine as markers of GFR [11]. Renal clearance of 10–30 kDa molecules appears selectively decreased in some individuals. In subsequent investigations of different populations, SPS has been associated with increased mortality or morbidity [13,14,15,16]. In a recent study comprising 2781 individuals, the adjusted risk of death at an eGFR_cystatin C_/eGFR_creatinine_ratio < 0.7 was considerably increased (hazard ratio (HR) 3.0, *p* < 0.001). In a subcohort of 567 subjects with normal measured GFR and no other diagnosis, the risk of all-cause mortality associated with SPS increased further (HR 7.3, *p* < 0.001) [17]. In an electron microscopy study of kidney biopsies from patients with diabetes mellitus, a thicker glomerular basement membrane (GBM) was associated with a lower eGFR_cystatin C_/eGFR_creatinine_ ratio. This association may be due to the shrinking pores caused by GBM thickening, which reduces the clearance of larger molecules [18]. In the Chronic Kidney Insufficiency Cohort (CRIC) study, a difference in eGFR_cystatin C_ and eGFR_creatinine_ of less than −15 mL/min/1.73 m^2^ was associated with an approximately twofold increase in the risk of ESKD and mortality compared to patients with a difference between −15 and 15 mL/min/1.73 m^2^ [19]. For more detailed information, we refer to an excellent review of this subject published recently [20].

One of the most robust predictive models for progressive CKD based on clinical markers is the Kidney Failure Risk Equation (KFRE) developed in a Canadian cohort [21]. Its accuracy was assessed in a population of 721,357 patients from 31 cohorts worldwide. The main outcome was kidney failure, which was defined as treatment with dialysis or kidney transplantation. An equation comprised of four variables (age, sex, eGFR, and UACR) and four additional variables (+ calcium, phosphate, bicarbonate, albumin) showed comparable performance, although evaluation based on the eight-variable equation was slightly superior. These equations are widely used through electronic applications (e.g., https://qxmd.com/calculate/calculator_308/kidney-failure-risk-equation-4-variable (accessed on 10 January 2024)), facilitating their integration into clinical practice. However, the risk of kidney failure may be overestimated, and a calibration factor may be necessary.

### 2.2. Blood Biomarkers

Elevated baseline serum levels of Kidney Injury Molecule-1 (KIM-1), a marker of tubular injury, have been shown to be strongly associated with the risk of progressive kidney function decline, regardless of other baseline clinical characteristics [22,23,24].

Neutrophil gelatinase-associated lipocalin (NGAL), a 25 kDa protein produced by the nephron epithelium, was identified as a biomarker for tubular injury and found to increase in patients with CKD. In a study of 112 patients with CKD, increased levels of NGAL (>107.8 ng/mL) predicted a significantly higher decrease in eGFR in CKD stages 1 or 2 (*p* < 0.0001). There was no difference in the change in eGFR between patients with CKD stages 3 to 5 [25]. In another study, based on the analysis of 126 CKD patients from outpatient clinics, NGAL was positively associated with the time to the composite renal and mortality endpoint (adjusted HR 2.7, *p* = 0.012) [26].

In two Swedish community-based cohorts of elderly individuals, higher serum soluble tumor necrosis factor receptor 1 (sTNFR1) was associated with an increased risk of progressing to a lower eGFR category in both cohorts (odds ratio (OR) 1.24–1.73 per standard deviation (SD) increment in sTNFR1, depending on the cohort and model applied *p* < 0.001), in longitudinal multivariable logistic regression models adjusted for inflammatory markers and established cardiovascular risk factors [27]. In a subgroup of individuals with GFR > 60 mL/min/1.73 m^2^ at baseline, higher sTNFR1levels were associated with incident CKD after 5 years in both cohorts (OR 1.46–2.04 per SD increment in sTNFR1 depending on cohort and model applied, *p* < 0.001).

C-reactive protein (CRP) has been demonstrated to be a risk factor not only for cardiovascular disease but also for CKD [28]. In type 1 diabetes, CRP level is related to CKD progression and has been suggested to predict disease severity [29]. In a study by Martin et al., CRP was shown to be independent predictor of renal composite endpoint (adjusted HR 1.4, *p* = 0.015) [26].

In the CRIC study, 899 of 3430 participants with CKD reached the composite endpoint of a >50% decline in eGFR or onset of ESKD. The association of plasma levels of interleukin 1 (IL-1), interleukin 1 receptor antagonist (IL-1RA), interleukin 6 (IL-6), tumor necrosis factor alpha (TNF-alpha), transforming growth factor β (TGFβ), CRP, fibrinogen, and serum albumin with progression was examined. Elevated levels of fibrinogen and TNF-alpha and decreased serum albumin were associated with a rapid loss of kidney function [30].

Galectin-3 (Gal-3) is a galactoside-binding protein involved in inflammation, angiogenesis, and organ fibrogenesis [31]. The link between plasma Gal-3 levels and the risk of developing CKD was investigated in 9148 participants in the Atherosclerosis Risk in Communities study [32]. After adjusting for confounding factors, elevated plasma Gal-3 levels were associated with an increased risk of developing new-onset CKD (quartile 4 vs. 1 HR 1.75, *p* < 0.001). In another study, the association between circulating levels of Gal-3 and CKD progression was investigated in a pooled study of 841 patients with CKD. In the adjusted analysis, doubling of Gal-3 was associated with an increased risk of CKD progression (HR 1.38, *p* = 0.044) [33].

### 2.3. Urine Biomarkers

In addition to serum KIM-1, urinary KIM-1 was also shown to predict CKD progression [34,35]. In two recent studies, Bienaime et al. and Greenberg et al. showed that a high level of urinary KIM-1 is significantly associated with a higher risk of CKD progression.

A biomarker that received substantial attention is Dickkopf-related protein 3 (DKK3), a secreted glycoprotein synthesized by stressed tubular epithelia, significantly expressed in mesenchymal progenitor and mesenchymal cells in vitro [36,37]. DKK3 is a profibrotic molecule that promotes tubulointerstitial fibrosis in experimental CKD models through modulation of the canonical Wnt/β-catenin signaling pathway [38]. Urinary DKK3 was found to be significantly elevated in patients with CKD (*n* = 575) compared to the general population (*n* = 481) (DKK3/creatinine ratio 431 pg/mg vs. 33 pg/mg) [39]. In the CKD population, DKK3 was associated with the female sex, lower body mass index (BMI), lower eGFR, and higher albuminuria. Patients with DKK3 levels < 200 pg/mg creatinine displayed no relevant reduction in eGFR. In contrast, eGFR declined in patients with urinary DKK3 levels exceeding 1000 pg/mg creatinine. Baseline urinary DKK3 concentrations of >4000 pg/mg creatinine were associated with a mean eGFR decline of 7.6% in the subsequent 12 months. DKK3 remained a significant predictor of eGFR decline after adjusting for multiple variables, including eGFR and UACR at baseline (*p* < 0.01). Higher urinary DKK3 levels were associated with a significantly greater loss of eGFR within all categories of albuminuria and were independent of the underlying kidney disease. These results were confirmed by Sanchez-Alamo et al. (*n* = 351) [40], who demonstrated that urinary DKK3 identified patients at high risk of CKD progression regardless of the cause of kidney injury. When DKK3 was added to a model comprising the eight-variable KFRE [41], the predictive power significantly increased (*p* < 0.01) [39]. The DKK3 test is commercially available.

### 2.4. Multi-Marker Models and Classifiers

Owens et al. [42] developed a biomarker panel to detect the risk of progressive CKD, defined as a >30% decrease in eGFR, dialysis initiation, or kidney transplantation. Ten of the thirty-seven potential baseline markers, selected based on pathophysiological processes, differed between patients with progressive and non-progressive CKD. These included serum creatinine, urea, tissue factor, osteopontin, sTNFR1, sTNFR2, tryptase, stem cell factor (SCF), and bicarbonate. All factors, except bicarbonate, were upregulated in progressive patients. A step-backward approach selected a biomarker panel consisting of bicarbonate, osteopontin, SCF, tissue factor, tryptase, urea, creatinine, and eGFR, which predicted progressive CKD with an accuracy of 84.3%, superior to that of serum creatinine, eGFR, albumin, and UACR, which achieved a cumulative predictive accuracy of only 75.8%.

Investigating serum and urinary markers in parallel, Colombo et al. [43] tested a panel of nine serum and thirteen urinary markers in 1629 individuals with type 1 diabetes and an eGFR > 30 mL/min/1.73 m^2^ at baseline with a follow-up of 5.1 years. Serum markers TNFR1, KIM-1, CD27 antigen, alpha1-microglobulin, syndecan 1, thrombomodulin, ratio of urinary epidermal growth factor (EGF)/monocyte chemoattractant protein-1 (MCP-1), and its individual components (EGF and MCP-1) were associated with follow-up eGFR in models adjusted for baseline eGFR and UACR. Serum TNFR1 and KIM-1 showed the strongest associations. Bienaime et al. tested 24 candidate biomarkers in the context of CKD progression and selected 16 for further investigation [34]. A combination of C-C motif chemokine ligand 2 (CCL2), EGF, KIM-1, NGAL, and TGF-alpha improved the prediction of eGFR decline compared to KFRE in a training set of 229 subjects. Unfortunately, validation in an independent cohort has not yet been attempted. A similar approach combining serum biomarkers TNFR1, NGAL, and complement C3a with clinical variables improved the prediction of CKD progression and mortality in a cohort of 139 CKD patients, unfortunately also without any validation [26]. Other similar studies [42] also reported increased accuracy of models based on a combination of multiple biomarkers, with a lack of demonstration of validity in independent samples. Overall, these studies support the value of combining multiple biomarkers into a model or classifier that demonstrates increased accuracy. This is likely because it incorporates multiple relevant pathways while simultaneously reducing the variability of single biomarkers [44].

Good et al. applied capillary electrophoresis coupled with mass spectrometry (CE-MS) to develop a high-dimensional urinary biomarker pattern, composed of 273 peptides (CKD273), associated with overt kidney damage in 230 subjects with various forms of CKD compared to 379 controls [45]. The most prominent changes upon which this classifier is based include a reduction in multiple collagen alpha-1(I) (COL1A1) (and also other collagen-derived) peptides and an increase in alpha-1-antitrypsin peptides. The authors hypothesized that these changes may reflect the attenuation of physiological collagen degradation, resulting in a reduced abundance of collagen peptides. This is accompanied by an increase in (chronic) inflammation, as reflected by the increased abundance of several alpha-1-antitrypsin peptides. In several retrospective studies, this proteomic classifier identified people at risk of diabetic kidney disease and progression in the microalbuminuria class earlier than the indices currently used in clinical practice [46,47,48]. Pontillo et al. [49] and Rodriguez-Ortiz et al. [50] tested the predictive power of the classifier CKD273 in various strata of eGFR, ranging from >80 to <30 mL/min/1.73 m^2^. In a study by Pontillo et al. [49], 2672 patients were included, of whom 394 were rapid progressors. In the eGFR strata 70–79 and >80 mL/min/1.73 m^2^, CKD273 outperformed UACR in predicting progression, whereas at an eGFR < 50, the area under the curve (AUC) for albuminuria was superior. In a study by Rodriguez-Ortiz et al. [50], the creation of subclassifiers of CKD273 for the various eGFR strata led to enhanced prediction of progression risk. The predictive accuracy surpassed that of albuminuria for patients with eGFR > 60 mL/min/1.73 m^2^. The performance of CKD273 subclassifiers was also significantly better than that of the KFRE at two and five years.

These data resulted in the initiation of the PRIORITY trial (NCT02040441), a prospective study in patients with type 2 diabetes. The trial aimed to test whether CKD273 is associated with the development of persistent microalbuminuria and to determine whether intervention with spironolactone reduces the increased risk of developing microalbuminuria in people with a high-risk CKD273 pattern compared with placebo [51]. Progression to microalbuminuria was observed in 28% of 216 high-risk participants and 9% of 1559 low-risk participants (HR 2.48, *p* < 0.0001) after adjustment for multiple baseline variables. A >30% decrease in eGFR from baseline was observed in 19% of high-risk and 4% of low-risk participants (HR 5.15, *p* < 0.0001). The CKD273 classifier identified patients with type 2 diabetes at risk of developing diabetic kidney disease earlier than conventional markers (eGFR and microalbuminuria), independent of clinical characteristics, in a prospective manner. However, spironolactone did not prevent disease progression in high-risk patients [52]. CKD273, acknowledged by the U.S. Food and Drug Administration with a letterofsupport (https://fda.report/media/99837/Biomarker-Letter-of-Support--June-14--2016--Harald-Mischak.pdf (accessed on 10 January 2024)), is the only novel biomarker demonstrating value in a prospective trial. It can serve as a very early risk predictor for deteriorating kidney function, guiding intervention at very early time points of the disease, before CKD (eGFR < 60 mL/min/1.73 m^2^ and/or microalbuminuria) is manifest. The CKD273 test is commercially available.

The reviewed non-invasive biomarkers considered for the detection of CKD and evaluation of the risk of progression are outlined in Appendix A.

While early diagnosis and prognosis of the progression of CKD are certainly relevant, the most valuable guidance appears to be the prediction of treatment response, especially in light of various potentially beneficial intervention methods available. To this end, the Lambers-Heerspink group developed a Parameter Response Efficacy (PRE) score to translate short-term drug effects into predictions of long-term effects on clinical outcomes [53]. The score is based on systolic blood pressure, UACR, serum potassium, hemoglobin, serum uric acid, blood glucose, total cholesterol, and BMI and was shown to predict drug efficacy on cardio-renal outcomes. However, no evidence demonstrating its value in guiding personalized intervention on an individual level is available yet. In a small study, Lindhardt et al. demonstrated the potential predictive value of CKD273 in the context of response to spironolactone [54]; however, the validation of these results in a larger study is still lacking. In a recent study, Jaimes Campos et al. [55] demonstrated the value of urinary peptides in predicting response to anti-hypertensive treatment. The authors identified 225 urinary peptides that were significantly associated with treatment response in a cohort from the PROVALID study [56]. A classifier based on 189 of these peptides (DKDp189) showed significant predictive value in two independent cohorts from the DIRECT-Protect 2 [57] and PRIORITY studies [52]. The most prominent peptides in this classifier were COL1A1 peptides, with reduced levels associated with a lack of response to renin-angiotensin system inhibitors. Additionally, peptides derived from alpha-1-antitrypsin showed increased levels indicating a reduced response. Interestingly, the classifier showed no significant predictive value in subjects not receiving anti-hypertensive treatment, strongly suggesting that it specifically predicts response to treatment. These results were recently taken one step further towards guiding intervention based on predicted impact of treatment in silico (depicted in Figure 2) investigated in 5585 individuals [58]. While an evident shortcoming of the approach presented is that the study was based on in silico intervention only, the large number of subjects/datasets included strongly supports this approach towards personalized intervention in CKD. Appendix A lists all biomarkers associated with drug prediction described in this section.

## 3. Non-Invasive Biomarkers to Estimate the Degree of Fibrosis

Fibrosis is a deregulated physiological response to tissue damage aimed at initiating repair processes and preserving tissue architecture and functionality (Figure 3) [59]. In the kidney, among others, nephron loss, microvascular damage, metabolic changes, oxidative stress, and inflammation result in fibrosis [59]. Dysregulated or excessive fibrogenesis leads to irreversible kidney damage due to parenchymal loss and replacement of functional with fibrotic tissue through extensive accumulation of the extracellular matrix (ECM) [60]. The epithelial-to-mesenchymal transition plays a crucial role in ECM accumulation and fibrosis progression in CKD [61]. The exact pathomechanisms of fibrogenesis through the fibrogenic niche [62] are currently being investigated, possibly enabling the identification of novel candidates for targeted therapy in CKD [63].

KF is assessed based on histopathological analysis of a kidney biopsy specimen. As KF is one of the best predictors of CKD progression, there is a need for non-invasive biomarkers that reflect the level of tubulointerstitial fibrosis. Due to its invasive nature, kidney biopsy is generally restricted to singular execution and is rarely performed repeatedly. Multiple tissue biomarkers are associated with KF and several functional pathways have been described. However, no routinely applicable non-invasive biomarkers for predicting the degree of KF are currently available.

### 3.1. Blood Biomarkers

Human epididymis protein-4 (HE4) is a putative serine protease inhibitor that has been proposed as a biomarker of ovarian cancer. *HE4* is upregulated in fibrosis-associated myofibroblasts and can suppress multiple serine proteases and matrix metalloproteinases (MMPs) in the kidney, inhibiting their ability to degrade collagen [65]. Neutralizing antibodies targeting HE4 enhance collagen degradation and inhibit fibrogenesis in different murine models [65]. Additionally, HE4 was found to be upregulated in human fibrotic kidneys, showing interstitial and tubular expression patterns, and serum concentrations of HE4 were significantly upregulated in CKD patients with biopsy-confirmed KF compared to healthy individuals [65]. Nagy et al. found significantly increased HE4 serum levels when comparing 113 female CKD patients (eGFR < 90 mL/min/1.73 m^2^) to 68 healthy controls [66]. Serum HE4 concentrations were higher in transplant recipients with reduced GFR, correlated with the degree of KF and tissue expression of HE4 within the biopsy [67].

Upregulated miR-21 expression was reported in the kidneys of mice with unilateral ureteral obstruction, a known mouse model for KF. Induction of fibrogenesis through TGFβ exposure in primary fibroblasts led to increased expression of miR-21, and ectopic expression of miR-21 in primary kidney fibroblasts induced myofibroblastic differentiation. Circulating miR-21 levels were significantly elevated in patients with allografts that showed a higher degree of IFTA. Moreover, miR-21 serum concentrations were independently associated with IFTA levels and eGFR (β = 0.307, *p* = 0.03 and β = −0.398, *p* = 0.006, respectively) [68].

WNT-1 inducible signaling pathway protein-1 (WISP-1) is an ECM protein associated with fibrogenesis in the kidney. WISP-1 was elevated in patients with diabetic nephropathy, Immunoglobulin A Nephropathy (IgAN), and focal segmental glomerular sclerosis compared to that in healthy subjects or patients with minimal-change disease. The authors found a significant correlation between serum WISP-1 levels and the biopsy-proven degree of KF (r = 0.475, *p* = 0.001) [69].

Lysyl oxidase (LOX) is involved in collagen cross-linking and its serum levels have been associated with fibrosis in the heart, lung, and liver [70]. Zhang et al. found increased serum LOX levels in 202 patients with KF compared to subjects without KF (*p* < 0.001). Additionally, LOX levels were higher in moderate–severe compared to mild KF cases (*p* < 0.001). Serum levels were associated with biopsy specimen fibrosis (r = 0.640, *p* < 0.001) [71].

SPARC-related modular calcium-binding protein 2 (SMOC2), cadherin-11 (CDH11), and pigment epithelium-derived factor (PEDF) were defined as potential fibrosis-associated biomarkers based on mouse single-cell RNA sequencing data. These biomarkers were validated in two cohorts (in both urine and serum) for their association with IFTA (*p* < 0.001 for each biomarker; *n* = 438 and for plasma biomarkers and *n* = 602 urinary biomarkers) [72].

### 3.2. Urine Biomarkers

Federico et al. studied urinary DKK3 excretion in mice, children, and adults with CKD compared to that in healthy individuals. DKK3 was detected in the urine of different fibrotic mouse models but was absent in healthy mice. To corroborate these findings, the group analyzed 72 pediatric and 36 adult patients and detected a correlation between urinary DKK3 and IF (AUC of 0.825) and TA (AUC of 0.864) [36].

TGFβ is a known key regulator of KF [73]. Investigation of TGFβ in 41 patients with idiopathic membranous glomerulonephritis (MGN), 25 healthy controls, and 6 kidney transplant recipients resulted in mixed findings. In MGN patients, the level of urinary TGFβ was significantly increased compared to that in healthy controls but did not correlate with the degree of fibrosis at the time of biopsy. However, urinary TGFβ collected one year prior to biopsy showed a highly significant correlation with the degree of future KF. The authors speculated that TGFβ might reflect ongoing renal inflammation and initial fibrotic changes, suggesting a role for TGFβ as a marker of the early stages of kidney injury and thus fibrogenesis [74]. The association between urinary TGFβ and the degree of kidney IF could not be confirmed in a small study of a heterogeneous group of nephropathies (fifteen cases of IgAN, nine of MGN, seven of rapid progressive glomerulonephritis (RPGN), eight of systemic lupus erythematodes (SLE), and nine of interstitial nephritis (IN)) [75].

Procollagen type III amino-terminal propeptide (PIIINP), a product of collagen III maturation and a potential indicator of fibrosis, was measured in the urine samples of 118 patients with CKD. PIIINP/creatinine ratios correlated with eGFR and were associated with IF (rho = 0.32, *p* = 0.0007) in corresponding kidney biopsies [76]. In 79 kidney transplant patients who underwent a 6-month protocol of follow-up biopsies, 24 h urine was assessed for PIIINP, TGFβ1, alpha-1-microglobulin, and albumin. Changes in creatinine clearance during a mean follow-up period of 4.3 years were also evaluated. Urinary PIIINP was significantly lower in patients with no KF than in those with mild–moderate IF assessed according to the Banff classification. Close correlations were found between urinary PIIINP and the degree of fibrosis (r = 0.410, *p* < 0.001) and between urinary excretion of TGFβ1 (r = 0.585, *p* < 0.001) and alpha-1-microglobulin (r = 0.438, *p* < 0.001). During the follow-up period, eGFR decline occurred in 42% of patients with urinary PIIINP/creatinine ratios >100 ng/mmol but in only 8% of patients with ratios below that threshold [77].

The utility of three biomarkers of collagen type III turnover (C3M, C3C, and PRO-C3) in assessing fibrotic burden in patients with IgAN was investigated in the serum and urine of 134 patients with IgAN [78]. Serum levels of PRO-C3 (rho = 0.31, *p* < 0.001) and C3C (rho = 0.23, *p* < 0.01) correlated positively with the degree of fibrosis in kidney biopsy, while urinary C3M/creatinine showed an inverse correlation (rho = −0.58, *p* < 0.0001); notably, urinary C3M/creatinine demonstrated the highest discriminative ability for advanced fibrosis (AUC of 0.81, *p* < 0.0001).

Matrix metalloproteinase-7 (MMP-7) is an endopeptidase and downstream transcriptional target of fibrosis-associated Wnt/β-catenin signaling. Urinary MMP-7 was elevated in 102 CKD patients compared to 20 healthy subjects (*p* = 0.03). Furthermore, in 30 of the 102 patients with available kidney biopsy specimens, MMP-7 levels were inversely correlated with eGFR (r = 0.304, *p* = 0.002) and associated with KF in the biopsy specimen (r = 0.635, *p* = 0.0016) [79].

Increased levels of urinary Gal-3 were found to be associated with lower eGFR, higher proteinuria levels, and with progression risk (*p* < 0.001) in a cohort of 280 patients. Logistic regression analysis revealed an association between urinary Gal-3 levels and higher degrees of IF (*p* = 0.003), TA (*p* = 0.002) and interstitial inflammation (*p* = 0.003) [80].

Urinary MCP-1 and collagen type IV, two proteins repeatedly reported to be involved in fibrogenesis in the kidney, were studied in 635 subjects. The study was primarily conducted on healthy donors of kidney allografts, with low levels of fibrosis among participants (79% had no fibrosis, 20% had 0–10% fibrosis, and 1% had fibrosis >10%). After adjusting for age, sex, and GFR, high urinary MCP-1, but not collagen type IV, was associated with IFTA (*p* = 0.0005) [81]. Interestingly, Wada et al. showed that MCP-1 blockade ameliorated KF by reducing collagen type I deposition and TGFβ expression in the unilateral ureteral obstruction mouse model [82].

Melchinger et al. investigated urinary uromodulin, a biomarker of tubular health exclusively produced in the thick ascending limb, in a cohort of 364 patients. Uromodulin was inversely associated with IFTA (*p* < 0.001) [83].

As evident from the data, several urinary molecules show a significant correlation with KF; however, they have frequently been investigated in relatively small cohorts and often in specific CKD etiologies. Many studies have been performed on transplant recipients, possibly not reflecting typical CKD patients. Most studies have been performed to support the potential pathomechanism found in animal experiments and to assess the potential clinical applicability of the investigated molecule. Potential biomarkers for KF appear to be either collagen fragments or proteins putatively involved in canonical fibrogenic pathways [84]. As fibrogenesis is a dynamic process involving several different pathways and one biomarker alone generally cannot reflect the different underlying mechanisms of fibrosis in various CKD etiologies, an approach combining multiple different proteins and peptide fragments in a model may solve the aforementioned dilemma.

### 3.3. Multi-Marker Models and Classifiers

CE-MS analysis of urinary peptides has proven successful in diagnosing different CKD etiologies and in risk stratification of CKD progression [52,85]. Consequently, Magalhaes et al. investigated the association between the urinary proteomic classifier CKD273 score and KF [86]. Forty-two kidney biopsies and urine samples were examined. CKD273 was significantly correlated with the degree of fibrosis (rho = 0.42, *p* = 0.0044). Interestingly, eGFR, albuminuria, and proteinuria were not significantly correlated with the degree of fibrosis in this cohort. Next, investigators aimed to identify fibrosis-associated peptides and found seven statistically significant ones, mainly collagen fragments corresponding to either type I or type III collagen (rho between −0.552 and −0.343, *p* < 0.05).

Catanese et al. investigated the urine proteome in the context of KF in 435 CKD patients with different etiologies. Most patients had biopsy-proven hypertensive ischemic nephropathy and IgAN. A urinary proteomic classifier, FPP_BH29, was created, containing 29 differentially excreted peptides associated with the degree of IFTA. Most of these peptides were derived from type I or III collagen, confirming the results of Magalhaes et al. [86]. A subset of 200 patients was used as the training cohort for classifier generation, while 235 patients were used for independent classifier testing. The patients were divided into two groups: low fibrosis (IFTA < 10%) and high fibrosis (IFTA > 15%). With an AUC of 0.84, the FPP_BH29 classifier demonstrated very good performance in the independent test cohort (*n* = 235). The application of FPP_BH29 to all test patients showed a highly significant correlation between FPP_BH29 and the degree of IFTA (rho = 0.496, *p* < 0.0001) [87].

In conclusion, the most advanced KF-predicting biomarkers that demonstrate sufficient clinical implementation evidence are serum HE4, urinary DKK3, PIIINP, and the proteomic classifier FPP_BH29. The maturity of these biomarkers supports their application for the non-invasive assessment of KF. All discussed biomarkers are listed in Appendix A.

## 4. Current Status and Future Directions

A substantial number of biomarkers is available, with sufficient associated evidence to support their use in assessing early-stage CKD, in the prognosis of progression, and in the non-invasive estimation of the degree of fibrosis. The most mature biomarkers are listed in Table 1.

For clinical application, a defined cut-off is required, which is unfortunately not provided in most cases. Furthermore, adjustment for relevant covariables is not always available and is inconsistent between studies. A more in-depth analysis of available data from cohorts where multiple different variables have been described in multiple publications (e.g., in the CRIC or the Chronic Kidney Disease in Children Study (CKiD) study) is highly desirable. For example, we identified 37 publications on different biomarkers in the CRIC study (using the search terms (“Chronic Renal Insufficiency Cohort”[Title/Abstract] OR CRIC[Title/Abstract]) AND (biomarker[Title/Abstract])), but no study has compared all of these different biomarkers, although there must be substantial overlap between the studies, which would allow for comparative assessment. Instead of publishing the association of the next biomarker with outcome, combining all available data and performing a comparative analysis of these different biomarkers in one cohort, or, better yet, in a prospective study, would be highly valuable. Another issue is the different performance of various platforms when analysing biomarkers, as demonstrated in the CKiD study for soluble urokinase-type plasminogen activator receptor [93].

The available literature strongly suggests that a holistic protein/proteomics-based approach holds promise to substantially improve the management of CKD, especially when used in conjunction with the current trend of personalized medicine. The blood and urinary biomarkers outlined in this review, combined with various genomic approaches (reviewed in [94]), will most likely further improve our ability to accurately diagnose various forms of kidney diseases and to predict the risk of progression and response to treatment. Combining biomarkers into panels instead of using single biomarkers shows promise, given the complexity of organisms and their exposure to environmental factors. Selecting biomarkers with statistical significance is crucial, and although high-dimensional approaches are better than linear combinations, overfitting can be a significant issue [95]. Therefore, validating any biomarker panel/model using independent datasets and avoiding overestimating performance (which also occurs when performing leave-one-out cross-validation) is essential. Multi-marker panels based on insignificant changes have no value and represent overfitting. The most appropriate approach is to use relevant statistics (including adjustment for multiple testing) and multidimensional algorithms (such as support vector machine), define optimal model parameters based on cross-validation, and then apply them in an independent test set.

Several diagnostic and prognostic tools available today are ready to be used and can support the diagnosis, prognosis, and selection of therapeutic strategies. Furthermore, several available biomarkers allow for the estimation of the degree of KF. Patients with kidney disease generally present to nephrologists with indicative clinical patterns, carrying the potential to enable definite diagnoses based on these scenarios, which can be augmented with non-invasive biomarker-guided diagnosis schemes.

Urinary peptidomics and proteomics have been applied extensively for several years in patients with CKD, demonstrating value in supporting earlier and more accurate detection, prognostic assessment, and prediction of response to treatment. They also promise a better understanding of kidney disease pathophysiology, and have been proposed as a “liquid biopsy” to discriminate various renal disorders [85,96]. Furthermore, as proteins are major drug targets, their analysis may allow for the evaluation of therapeutic efficacy at the protein signaling pathway level [97].

The literature also indicates the advantages of urine as a specimen when compared to plasma or serum, including reduced complexity, increased stability, increased representation of kidney-related information (as it is produced in the kidneys), and non-invasive repeat sampling in large amounts. Approximately 70% of the urinary proteome originates from the kidneys and therefore provides a good representation of their condition [98]. Blood-based biomarkers may offer a more systemic snapshot of the physiological state of the body, capturing the interactions between various organ systems. However, this systemic perspective could limit the specificity of biomarkers, as it might not reflect kidney damage. Among the novel biomarkers, CKD273 appears the most advanced. It has been validated in multiple longitudinal studies and its diagnostic, prognostic, and drug response prediction utility has been confirmed. Furthermore, a cost-efficiency analysis was performed for this classifier [1]. As a result, the application of this approach in patient management has been implemented in Germany, where the first health insurance companies are now reimbursing this test (https://www.berliner-zeitung.de/gesundheit-oekologie/gesetzliche-krankenkasse-zahlt-urintest-zur-krebs-praevention-li.2191412 (accessed on 28 February 2024)).

## 5. Conclusions

Collectively, a substantial number of biomarkers is available with robust evidence supporting their utility in early CKD assessment, progression prognosis, and non-invasive fibrosis estimation. The literature strongly advocates for a holistic protein/proteomics-based approach, synergized with genomic methods, to significantly enhance CKD management, particularly within the framework of personalized medicine. As a result of the currently available evidence, assessing the potential benefits, especially of biomarkers guiding intervention, should be a major goal of the scientific community. Additional studies aiming to identify potential biomarkers that may be of use in the context of CKD do not appear to be of the highest priority, and thorough testing of the currently available biomarkers seems more likely to result in effective and significant patient benefits.

## Figures and Tables

**Figure 1 ijms-25-03678-f001:**
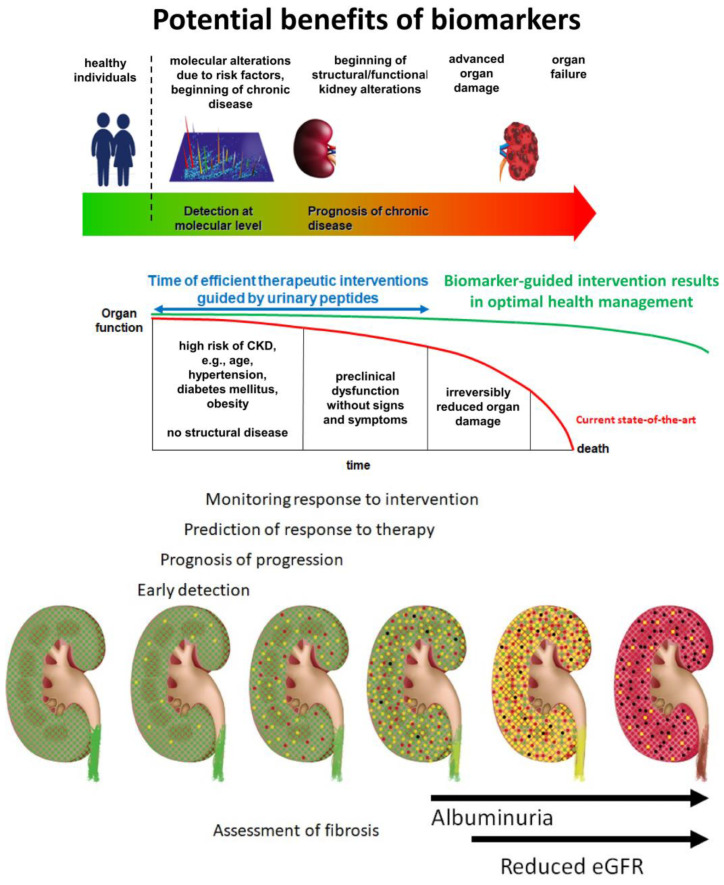
Graphical depiction of a model of chronic kidney disease (CKD) progression and the current needs and opportunities for improvement in patient management. In the lower panel, the progression of kidney disease is depicted and the specific context of biomarker use is indicated. Healthy nephrons (green) experience an impact that results in initial reversible damage (indicated in yellow). Without appropriate intervention, damage is expected to become irreversible (indicated in red), leading to destruction and replacement by fibrotic tissue (black). The upper panel indicates the potential benefits of applying molecular biomarkers. Ideally, these biomarkers should enable early detection and targeted intervention before irreversible organ damage occurs. Abbreviations: eGFR: estimated glomerular filtration rate.

**Figure 2 ijms-25-03678-f002:**
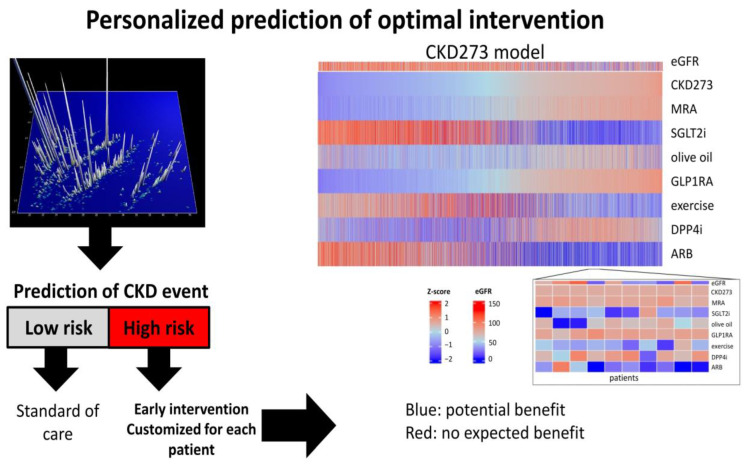
Biomarker-guided approach towards personalized intervention in chronic kidney disease (CKD). Biomarkers can be used to determine the risk of CKD onset and progression. In the case of increased risk, the impact of treatment with different drugs on biomarkers can be modeled based on previous data. Based on these models, optimal personalized intervention can be suggested [58]. Abbreviations: ARB: angiotensin receptor blockers; DPP4i: Dipeptidyl peptidase-4 inhibitor; eGFR: estimated glomerular filtration rate; GLP1RA: glucagon-like peptide 1 receptor agonist; MRA: mineralocorticoid receptor antagonist; SGLT2i: sodium-glucose cotransporter 2 inhibitor.

**Figure 3 ijms-25-03678-f003:**
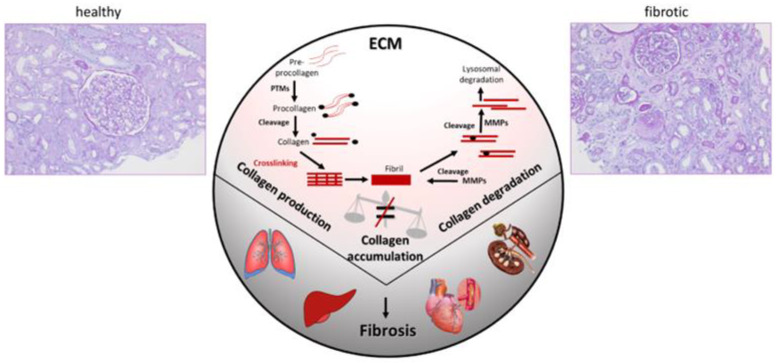
Fibrosis results from imbalanced homeostasis of extracellular matrix (ECM). While current efforts to optimize therapeutic intervention mostly target synthesis of ECM and, in many cases, different types of collagens, accumulating evidence suggests that attenuation of degradation may be a key factor driving fibrosis. The figure was adapted from Latosinska et al. [64]. Abbreviations: ECM: extracellular matrix; MMPs: matrix metalloproteinases.

**Table 1 ijms-25-03678-t001:** Biomarkers that have reached a level of maturity that sufficiently supports clinical use.

Intended Use	Biomarker	Findings	References
(1) Early diagnosis	CKD273	Enables detection of CKD at early stage, superior to albuminuria or eGFR	[46,49,50,52,88]
(2) Assessment of fibrosis	FPP_29BH	Detection of fibrosis	[87]
	C3M	Association with fibrosis	[78]
	HE4	Increased in fibrosis	[66,67]
	LOX	Increased in fibrosis	[71]
	DKK3	Increased in fibrosis	[36]
(3) Prognosis of progression	KIM-1	Increased in progressive CKD	[23,24,34,35]
	NGAL	Increased in progressive CKD	[26,34]
	TNFR1	Increased in progressive CKD	[24,26,35,89,90]
	TNFR2	Increased in progressive CKD	[23,24,35,89,90]
	EGF	Reduced levels indicate progression	[34,35,91]
	MCP-1	Increased in progressive CKD	[35]
	DKK3	Increased in progressive CKD	[39,40]
	CKD273	Increased in progressive CKD	[46,47,52,88,92]
(4) Prediction of drug response	DKDp189	Prediction of response to anti-hypertensive treatment	[55]

Abbreviations: AUC: area under the curve; CKD: chronic kidney disease; DKK3: Dickkopf-related protein 3; eGFR: estimated glomerular filtration rate; EGF: epidermal growth factor; HE4: human epididymis protein-4; KIM-1: Kidney Injury Molecule-1; LOX: lysosyl oxidase; MCP-1: monocyte chemoattractant protein; NGAL: neutrophil gelatinase-associated lipocalin; TNFR: tumor necrosis factor receptor.

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
