# Peer review of "Assessment and Risk Prediction of Chronic Kidney Disease and Kidney Fibrosis Using Non-Invasive Biomarkers"

_ijms, 2024, doi:10.3390/ijms25073678_

Round 1

Reviewer 1 Report

Comments and Suggestions for Authors

The article is valuable in its effort to provide a summary of the present evidences about non-invasive biomarkers for chronic kidney disease and to try to guide the reader to understand those with clinical usefulness. However different points need an improvement and so extensive revisions are needed.

Major revisions

First of all, the methods used for conducting this review are lacking. Some of them are anticipated in the abstract (“We specifically focus on …. including at least 100 subjects”), but then they are not reported in the text of the article. At the same time other pieces of methods are written in different section of the article (i.e in the introduction lines 69-70, in the conclusion lines 444-446). Even if this is not a systematic review, it is necessary to introduce a paragraph or a section which states clearly the methods used for the review, removing at the same time the pieces of information distributed throughout the article.

The authors stated that “focus is placed on urinary biomarkers”, however the article is almost equally divided between urinary and blood biomarkers: please explain or correct the sentence about the focus of the article.

Paragraph 2: lines 88-97 present a repetition of what is already written in the introduction: please amend. Moreover AUCR is both a consequence of kidney disease and a cause of damage, in fact one of the aims of therapeutic intervention is to reduce the amount of urinary protein: please explain.

Section 2.1: the section is about routine clinical markers, the authors stated that the conventional biomarkers are eGFR and AUCR, the first is described in the section, but the description of the second is lacking. Please add studies about AUCR as a biomarkers. Moreover eGFR is described almost entirely linked to the eGFRcystatin C eGFRcreatinine ratio of studies about the “shrunken pore syndrome”: is this the only use of eGFR as a biomarkers in literature? The bibliography reports some studies about the “shrunken pore syndrome” but not the most recent ones published from 2021 to 2023: why? Please add missed studies.

As there are many studies in bibliography about NGAL (from 19 to 24), why only one is described in the text?

It would be useful to add a table (o some tables, ie blood biomarkers, urinary biomarkers, fibrosis, etc) in which all the studies considered in the review are summarized (ie: number of patients, brief results, type of patients considered, etc) divided for biomarker. So in the article, the authors could give a comprehensive view of results without reporting each study singularly.

It could be advisable to separate in a specific paragraph all the studies in which panels of various biomarkers were studied, because some of them included both serum and urinary biomarkers and other were a combination of new proteins and routinely used parameters, such as creatinine, urea and bicarbonate.

Paragraph 2.3: lines 237-243 refer to a study analyzing a score that combined different parameters: this is not in line with the title of the paragraph “urine biomarkers”: please amend.

Could some information about the urinary peptides included in CKD273, in the panel of PROVALID study and on DKDp189 be provided?

Lines 259-269: the content of the paragraph is not in line with the title of the paragraph in itself, it could be included in the conclusion of the manuscript or in a discussion paragraph as it contains general considerations.

Table 1 needs revision: probably it would be better to highlight the most important biomarkers in tables where all the studies considered are presented not in a separated table. Then, in the table there is a mixture of all type of biomarkers, some presented in only one study, in some of them the number of patients and the statistical analysis is reported in other ones not. It is a bit confused: please try to clarify or change the table.

Line 343 “several studies” investigated the potential of TGBβ as a biomarker, but only 3 of them are reported: why? Moreover all the studies reported had a number of patients <100, that was the cut-off stated by authors in the abstract to include a study: please explain.

Also for PIIINP one of the analyzed studies had less than 100 patients: please explain

Gal-3 “was investigated in multiple studies”: why was only one study reported/analyzed?

Lines 395-401 underline some weaknesses of the studies previously reported, but also studies reported in other paragraphs consider only a specific CKD etiology, ie diabetes mellitus, or patients with a prevalence of IgAN and ischemic nephropathy (418-419), or patients with CKD of unspecified etiologies: to comment on this point the inclusion and exclusion criteria of the studies included in the review should be clearly stated.

Conclusion should be more concise underlining the most important points of the review and eventually briefly suggesting future directions.

Minor revisions

The article contains some typos to be amended, ie: line 43 “in this context: They”, line 183 “(MCP-1), MCP-1,”, line 346 “in mixed findings: In” line 405 “CKD etiologies. An”: please revise the whole text accurately.

The text contains a lot of acronyms, not all of them have the corresponding name, some have it more than once, ie: IFTA line 58 and line 341, line 332: SMOC2, PEDF, SERPINF1; line 337: DKK3. Please revise and consider to add a section at the end with all acronyms explained, this could make the text easily readable.

Figure 1 caption: lines 82-85 are a repetition of introduction, not an explication of the figure.

Comments on the Quality of English Language

Line 195 the adjective “outstanding” could be changed with “lacking” or something similar, as outstanding usually means “extremely good, excellent"

Author Response

Response to Reviewer 1 Comments

General comment: The article is valuable in its effort to provide a summary of the present evidences about non-invasive biomarkers for chronic kidney disease and to try to guide the reader to understand those with clinical usefulness. However different points need an improvement and so extensive revisions are needed.

Response: We appreciate the Reviewer's valuable feedback on our manuscript. We have taken the feedback into account and made the necessary revisions to improve the quality of the manuscript.

Major revisions

Comment 1: First of all, the methods used for conducting this review are lacking. Some of them are anticipated in the abstract (“We specifically focus on …. including at least 100 subjects”), but then they are not reported in the text of the article. At the same time other pieces of methods are written in different section of the article (i.e in the introduction lines 69-70, in the conclusion lines 444-446). Even if this is not a systematic review, it is necessary to introduce a paragraph or a section which states clearly the methods used for the review, removing at the same time the pieces of information distributed throughout the article.

Response: Thank you for bringing this to our attention. We have now included a description of the search strategy used.

Comment 2: The authors stated that “focus is placed on urinary biomarkers”, however the article is almost equally divided between urinary and blood biomarkers: please explain or correct the sentence about the focus of the article.

Response: Thank you for raising this valid point. We agree with the Reviewer and have omitted this statement, as the review covers non-invasive biomarkers measured in urine and blood.

Comment 3: Paragraph 2: lines 88-97 present a repetition of what is already written in the introduction: please amend. Moreover, AUCR is both a consequence of kidney disease and a cause of damage, in fact one of the aims of therapeutic intervention is to reduce the amount of urinary protein: please explain.

Response: We agree with the Reviewer and have deleted this paragraph as it essentially duplicates content from the introduction.

Comment 4: Section 2.1: the section is about routine clinical markers, the authors stated that the conventional biomarkers are eGFR and AUCR, the first is described in the section, but the description of the second is lacking. Please add studies about AUCR as a biomarkers. Moreover eGFR is described almost entirely linked to the eGFRcystatin C eGFRcreatinine ratio of studies about the “shrunken pore syndrome”: is this the only use of eGFR as a biomarkers in literature? The bibliography reports some studies about the “shrunken pore syndrome” but not the most recent ones published from 2021 to 2023: why? Please add missed studies.

Response: Thank you for bringing this point to our attention. To avoid potential misunderstandings, we now more clearly state that UACR and eGFR based on serum creatinine are routinely used for assessment of kidney function. UACR (we are sorry for the incorrect acronym) is widely used and no alternative is available to the determination of urinary albumin, hence we felt a long discussion of UACR is of little added value. However, we added the main concern and shortcoming: the very high biological variability. This is different for eGFR estimates, where alternatives to measuring creatinine are available. Based on the suggested by the Reviewer, we further added a very recent and extensive review on the shrunken pore syndrome.

Comment 5: As there are many studies in bibliography about NGAL (from 19 to 24), why only one is described in the text?

Response: There was no specific reason, we felt the other studies did not contribute a lot in addition. We agree with the Reviewer that this is unclear and have therefore added some information on other studies in the text.

Comment 6: It would be useful to add a table (o some tables, ie blood biomarkers, urinary biomarkers, fibrosis, etc) in which all the studies considered in the review are summarized (ie: number of patients, brief results, type of patients considered, etc) divided for biomarker. So in the article, the authors could give a comprehensive view of results without reporting each study singularly.

Response: We thank the Reviewer for this valid suggestion. We have tried, but, mostly due to the heterogeneity of the study designs (different aims, endpoints, variables assessed and reported, etc.), we were unable to generate a sufficiently homogenous table that also would comprehensively represent the information. We therefore opted for reporting the studies in the text. However, we have now added supplementary tables outlining the main study characteristics including biomarker and aim, number of subjects, and some measure of performance.

Comment 7: It could be advisable to separate in a specific paragraph all the studies in which panels of various biomarkers were studied, because some of them included both serum and urinary biomarkers and other were a combination of new proteins and routinely used parameters, such as creatinine, urea and bicarbonate.

Response: Based on the very appropriate suggestion, we have reorganized the subsections and now also present a section on multi-marker approaches.

Comment 8: Paragraph 2.3: lines 237-243 refer to a study analyzing a score that combined different parameters: this is not in line with the title of the paragraph “urine biomarkers”: please amend.

Response: Thank you very much for this comment. As indicated above, we have now re-organized the subsections.

Comment 9: Could some information about the urinary peptides included in CKD273, in the panel of PROVALID study and on DKDp189 be provided?

Response: We are grateful for this comment and now provide some detailed information on the most prominent (based on the changes observed in disease) biomarkers contained in these classifiers.

Comment 10: Lines 259-269: the content of the paragraph is not in line with the title of the paragraph in itself, it could be included in the conclusion of the manuscript or in a discussion paragraph as it contains general considerations.

Response: We fully agree with the Reviewer. We have now removed this paragraph and included its content in the new section “Current status and future directions”, at the same time trying to avoid any duplication.

Comment 11: Table 1 needs revision: probably it would be better to highlight the most important biomarkers in tables where all the studies considered are presented not in a separated table. Then, in the table there is a mixture of all type of biomarkers, some presented in only one study, in some of them the number of patients and the statistical analysis is reported in other ones not. It is a bit confused: please try to clarify or change the table.

Response: Thank you for pointing this out. We would prefer to leave this this table in, as it gives the overview of most important biomarkers. Moreover, as suggested by the Reviewer, we have included more detailed information about individual studies and biomarkers in separate supplementary tables.

Comment 12: Line 343 “several studies” investigated the potential of TGBβ as a biomarker, but only 3 of them are reported: why? Moreover, all the studies reported had a number of patients <100, that was the cut-off stated by authors in the abstract to include a study: please explain.

Response: Thank you for the comment. We adhered to the 100 subject restriction when selecting the “basis” for the manuscript, but on several occasions also added additional studies with less than 100 subjects if helpful to support the “basis findings”. It's worth noting that the association of TGFβ with kidney fibrosis has been documented in previous studies outside the search period. Our aim was to provide mainly an updated perspective on developments in the field. Thus, we chose to omit the statement " and several studies investigated its biomarker potential in the context of IFTA " to avoid confusion.

Comment 13: Also for PIIINP one of the analyzed studies had less than 100 patients: please explain

Response: Thank you for the comment. As mentioned above, we adhered to the 100 subject restriction when selecting the “basis” for the manuscript, but on several occasions also added additional studies with less than 100 subjects if helpful to support the “basis findings”.

Comment 14: Gal-3 “was investigated in multiple studies”: why was only one study reported/analyzed?

Response: Thank you for pointing this out. The statements refer to the previous review article, PMID: 35328545, and the role of Gal-3 as a biomarker of kidney disease. We omitted this sentence to avoid confusion. Additional studies on Gal-3 were also included in the context of early detection (PMID: 28865675), and CKD progression (30596173).

Comment 15: Lines 395-401 underline some weaknesses of the studies previously reported, but also studies reported in other paragraphs consider only a specific CKD etiology, ie diabetes mellitus, or patients with a prevalence of IgAN and ischemic nephropathy (418-419), or patients with CKD of unspecified etiologies: to comment on this point the inclusion and exclusion criteria of the studies included in the review should be clearly stated.

Response: Thank you for this comment. We have now presented the search strategy in the introduction.

Comment 16: Conclusion should be more concise underlining the most important points of the review and eventually briefly suggesting future directions.

Response: We thank the Reviewer for the comment. The information initially included in the conclusion section was transferred to the separate paragraphs titled "Current status and future directions," and a new shorter and more concise conclusion paragraph was prepared.

Minor revisions

Comment 17: The article contains some typos to be amended, ie: line 43 “in this context: They”, line 183 “(MCP-1), MCP-1,”, line 346 “in mixed findings: In” line 405 “CKD etiologies. An”: please revise the whole text accurately.

Response: We have addressed the indicated issues and checked the text for typographical errors.

Comment 18: The text contains a lot of acronyms, not all of them have the corresponding name, some have it more than once, ie: IFTA line 58 and line 341, line 332: SMOC2, PEDF, SERPINF1; line 337: DKK3. Please revise and consider to add a section at the end with all acronyms explained, this could make the text easily readable.

Response: The acronyms were adjusted as suggested, and an additional section listing them was included.

Comment 19: Figure 1 caption: lines 82-85 are a repetition of introduction, not an explication of the figure.

Response: Thank you for this comment. We have revised the indicated part so that it better describes the content of the figure.

Comments on the Quality of English Language

Comment 20: Line 195 the adjective “outstanding” could be changed with “lacking” or something similar, as outstanding usually means “extremely good, excellent"

Response: Thank you for this suggestion. The text has been corrected.

Reviewer 2 Report

Comments and Suggestions for Authors

Comments on the article titled: Assessment and Risk Prediction of Chronic Kidney Diseases
and Kidney Fibrosis Using Non-Invasive Biomarkers by Harald D Rupprecht et al. submitted to mdpi International Journal of Molecular Sciences

This is a very fine article that highlights the importance of non-invasive biomarkers in assessing and managing chronic kidney diseases (CKD) and kidney fibrosis.

It is a literature review that does a fine job of organizing knowledge without adding any value.

Some key biomarkers discussed in the literature include serum markers such as TNFR1, KIM-1, CD27, alpha1-microglobulin, syndecan 1, thrombomodulin, as well as urinary markers like Epidermal growth factor (EGF), Monocyte chemoattractant protein-1 (MCP-1), NGAL, and TGF-alpha

The main findings indicate that these biomarkers have shown associations with eGFR decline and CKD progression, providing valuable insights for monitoring and predicting kidney disease outcomes. Non-invasive biomarkers play a crucial role in predicting the risk of developing kidney diseases by identifying patients at high progression risk. By utilizing biomarkers such as C-C motif chemokine ligand 2 (CCL2), EGF, KIM-1, NGAL, and TGF-alpha, researchers have been able to improve the prediction of eGFR decline and CKD progression. These biomarkers help in stratifying patients based on their risk profile, allowing for personalized treatment strategies and interventions to be implemented effectively.

The use of non-invasive biomarkers for early detection and management of kidney diseases has significant implications for clinical practice. These biomarkers enable healthcare providers to diagnose CKD at early stages, predict disease progression, assess the degree of kidney fibrosis, predict therapy response, and monitor the effects of therapeutic interventions

By focusing on urinary biomarkers due to their advantages in terms of sample collection and kidney-related information representation, clinicians can make informed decisions regarding patient care and treatment plans. Additionally, integrating multiple biomarkers in predictive models enhances the accuracy of risk assessment and disease progression monitoring, ultimately leading to improved patient outcomes.

Some of the parameters are considered consequences of kidney damage rather than indicators of the molecular changes driving CKD initiation and progression.

The Authors very accurately underscore the urgent need for sensitive and specific non-invasive biomarkers that can address the following key goals in CKD management: early detection of the disease, prediction of CKD progression risk, estimation of kidney fibrosis degree, prediction of therapy response, and monitoring therapeutic intervention effects

Suggestions:

The authors point to several mathematical models and classifiers that show significant predictive value in the assessment of chronic kidney disease. These mathematical models and classifiers play a key role in predicting disease progression and guiding personalized interventions for patients with kidney disease. Unfortunately, information on existing models is scattered throughout the text. I think it is the models and classifiers of the future that will play a key role in CKD risk assessment, especially with the rapid development of numerical methods and artificial intelligence. I believe that the creation of computational models will be crucial for future researchers. Therefore, I strongly suggest that an additional chapter on mathematical models and prediction using computational tools be added to the manuscript. Such a chapter is extremely important in the context of a title containing the word “prediction”.

Technical issues:

1.      The layout of References does not comply with Int. J. Mol. Sci. standard – see guidance for Authors

2.      Please pay attention to the usage of dashes and hyphens when quoting article numbers in brackets, see the example lines 143 (dash-proper) and 146 (hyphen-unproper)

3.      There are many sentences in the text that are too long, which makes them difficult to understand (see example lines 42-45)

4.      Lines 32-36 are basically the same sentences, please correct

Good Luck!

Comments on the Quality of English Language

Language basically fine; text contains many sentences which are too long and make it difficult to understand.

Author Response

Response to Reviewer 2 Comments

General Comments: Comments on the article titled: Assessment and Risk Prediction of Chronic Kidney Diseases and Kidney Fibrosis Using Non-Invasive Biomarkers by Harald D Rupprecht et al. submitted to mdpi International Journal of Molecular Sciences

This is a very fine article that highlights the importance of non-invasive biomarkers in assessing and managing chronic kidney diseases (CKD) and kidney fibrosis. It is a literature review that does a fine job of organizing knowledge without adding any value. Some key biomarkers discussed in the literature include serum markers such as TNFR1, KIM-1, CD27, alpha1-microglobulin, syndecan 1, thrombomodulin, as well as urinary markers like Epidermal growth factor (EGF), Monocyte chemoattractant protein-1 (MCP-1), NGAL, and TGF-alpha

The main findings indicate that these biomarkers have shown associations with eGFR decline and CKD progression, providing valuable insights for monitoring and predicting kidney disease outcomes. Non-invasive biomarkers play a crucial role in predicting the risk of developing kidney diseases by identifying patients at high progression risk. By utilizing biomarkers such as C-C motif chemokine ligand 2 (CCL2), EGF, KIM-1, NGAL, and TGF-alpha, researchers have been able to improve the prediction of eGFR decline and CKD progression. These biomarkers help in stratifying patients based on their risk profile, allowing for personalized treatment strategies and interventions to be implemented effectively.

The use of non-invasive biomarkers for early detection and management of kidney diseases has significant implications for clinical practice. These biomarkers enable healthcare providers to diagnose CKD at early stages, predict disease progression, assess the degree of kidney fibrosis, predict therapy response, and monitor the effects of therapeutic interventions

By focusing on urinary biomarkers due to their advantages in terms of sample collection and kidney-related information representation, clinicians can make informed decisions regarding patient care and treatment plans. Additionally, integrating multiple biomarkers in predictive models enhances the accuracy of risk assessment and disease progression monitoring, ultimately leading to improved patient outcomes.

Some of the parameters are considered consequences of kidney damage rather than indicators of the molecular changes driving CKD initiation and progression.

The Authors very accurately underscore the urgent need for sensitive and specific non-invasive biomarkers that can address the following key goals in CKD management: early detection of the disease, prediction of CKD progression risk, estimation of kidney fibrosis degree, prediction of therapy response, and monitoring therapeutic intervention effects

Response: We appreciate the Reviewer’s positive feedback on our article and valuable comments and suggestions, which helped us enhance the quality of our manuscript.

Suggestions:

Comment 1: The authors point to several mathematical models and classifiers that show significant predictive value in the assessment of chronic kidney disease. These mathematical models and classifiers play a key role in predicting disease progression and guiding personalized interventions for patients with kidney disease. Unfortunately, information on existing models is scattered throughout the text. I think it is the models and classifiers of the future that will play a key role in CKD risk assessment, especially with the rapid development of numerical methods and artificial intelligence. I believe that the creation of computational models will be crucial for future researchers. Therefore, I strongly suggest that an additional chapter on mathematical models and prediction using computational tools be added to the manuscript. Such a chapter is extremely important in the context of a title containing the word “prediction”.

Response: We appreciate the valuable suggestion and agree with the Reviewer regarding the significance of the biomarker panels. However, it is not within the scope of this review to provide a detailed description of the mathematical models and computational tools used for combining biomarkers. Since the topic is vast, we have decided to concentrate on the practical and general aspects of biomarker panel development, which have now been included in the "Current status and future directions."

Comment 2: Technical issues:

  1. The layout of References does not comply with Int. J. Mol. Sci. standard – see guidance for Authors
  2. Please pay attention to the usage of dashes and hyphens when quoting article numbers in brackets, see the example lines 143 (dash-proper) and 146 (hyphen-unproper)
  3. There are many sentences in the text that are too long, which makes them difficult to understand (see example lines 42-45)
  4. Lines 32-36 are basically the same sentences, please correct

Good Luck!

Response: We appreciate the Reviewer for thoroughly reviewing the manuscript. We have made several revisions to improve the quality of English throughout the text.

Comments on the Quality of English Language

Comment 3: Language basically fine; text contains many sentences which are too long and make it difficult to understand.

Response: We appreciate the Reviewer for thoroughly reviewing the manuscript. We have made several revisions to improve the quality of English throughout the text.

Round 2

Reviewer 1 Report

Comments and Suggestions for Authors

The authors addressed all the issues previously raised. The revised version is fine.

I thank the authors for their work.

Minor revision: the title of paragraph 3 is in the figure caption, please amend.

Author Response

Response to Reviewer 1 comments

General comment: The authors addressed all the issues previously raised. The revised version is fine. I thank the authors for their work.

Response: We appreciate the Reviewer's thorough evaluation of our manuscript and the constructive comments provided, which have significantly contributed to its improvement. We are pleased to hear that we have successfully addressed all the comments raised.

Minor revision:

Comment 1: the title of paragraph 3 is in the figure caption, please amend.

Response: Thank you for bringing this to our attention. We have now relocated the title of paragraph 3 to a separate line, as suggested.
